# Genome-Wide Identification of the Maize Chitinase Gene Family and Analysis of Its Response to Biotic and Abiotic Stresses

**DOI:** 10.3390/genes15101327

**Published:** 2024-10-15

**Authors:** Tonghan Wang, Changjin Wang, Yang Liu, Kunliang Zou, Minghui Guan, Yutong Wu, Shutong Yue, Ying Hu, Haibing Yu, Kaijing Zhang, Degong Wu, Junli Du

**Affiliations:** 1College of Agriculture, Anhui Science and Technology University, Fengyang 233100, China; 18767401521@163.com (T.W.); wangcj@ahstu.edu.cn (C.W.); zkl151231@163.com (K.Z.); wuyutong1385@163.com (Y.W.); y19855009352@163.com (S.Y.); huying2191@163.com (Y.H.); yuhb@ahstu.edu.cn (H.Y.); kaijing.zhang@gmail.com (K.Z.); 2College of Resource and Environment, Anhui Science and Technology University, Fengyang 233100, China; 18298115131@163.com (Y.L.); mhbelucky@163.com (M.G.); 3Anhui Province International Joint Research Center of Forage Bio-Breeding, Chuzhou 233100, China

**Keywords:** maize, Chitinase, gene family, abiotic and biotic stresses

## Abstract

Background/Objectives: Chitinases, enzymes belonging to the glycoside hydrolase family, play a crucial role in plant growth and stress response by hydrolyzing chitin, a natural polymer found in fungal cell walls. This study aimed to identify and analyze the maize chitinase gene family, assessing their response to various biotic and abiotic stresses to understand their potential role in plant defense mechanisms and stress tolerance. Methods: We employed bioinformatics tools to identify 43 chitinase genes in the maize B73_V5 genome. These genes were characterized for their chromosomal positions, gene and protein structures, phylogenetic relationships, functional enrichment, and collinearity. Based on previous RNA-seq data, the analysis assessed the expression patterns of these genes at different developmental stages and under multiple stress conditions. Results: The identified chitinase genes were unevenly distributed across maize chromosomes with a history of tandem duplications contributing to their divergence. The ZmChi protein family was predominantly hydrophilic and localized mainly in chloroplasts. Expression analysis revealed that certain chitinase genes were highly expressed at specific developmental stages and in response to various stresses, with ZmChi31 showing significant responsiveness to 11 different abiotic and biotic stresses. Conclusions: This study provides new insights into the role of chitinase genes in maize stress response, establishing a theoretical framework for exploring the molecular basis of maize stress tolerance. The identification of stress-responsive chitinase genes, particularly ZmChi31, offers potential candidates for further study in enhancing maize resistance to environmental challenges.

## 1. Introduction

Chitinase (EC 3.2.1.14), belongs to the glycoside hydrolase (GH) family, is prevalent in microorganisms [1,2,3], plants and animals that facilitates the breakdown of natural polymers such as chitin, chitosan, lipochitooligosaccharides, peptidoglycan, and glycoproteins that contain N-acetylglucosamine [4,5,6,7]. Plant chitinases are classified into seven groups (I to VII) according to their amino acid compositions [8], with groups I, II, IV, VI, and VII being part of glycoside hydrolase family 19 (GH19), which are commonly observed in higher plants [9,10]. Conversely, Classes III and V chitinases are classified under glycoside hydrolase family 18 (GH18), exhibiting a broad distribution across diverse organisms, including plants, animals, fungi, bacteria, and viruses [11,12]. Chitinases, which are enzymes characterized by their physicochemical traits and catalytic functions, are members of the glycoside hydrolase superfamily as indicated by the InterPro identifier IPR017853 [13,14]. This superfamily encompasses two distinct types of chitin-degrading enzymes: endo-chitinases and exo-chitinases. The Enzyme Commission (EC) number for endo-chitinases 3.2.1.14 indicates that the enzyme is enzymatically recognized as such, exhibit a random mode of action, cleaving the glycosidic bonds within the chitin polymer at various points along the chain. These enzymes are primarily associated with two specific glycoside hydrolase families: GH18 and GH19 [15]. Endo-chitinases from the GH18 family are notably ubiquitous across various biological systems, whereas those from the GH19 family are primarily associated with plants and Streptomyces bacteria [16,17,18]. In contrast, exo-chitinases, also referred to as chitobiosidases (EC 3.2.1.52), function processively by sequentially removing N-acetylglucosamine (GlcNAc) units from the non-reducing end of the chitin chain. These enzymes are categorized under the glycoside hydrolase family GH20 [19,20,21]. Chitinase activity has been detected in *Triticum aestivum* [22], *Arabidopsis thaliana* [23], *Glycine max* [24], *Sorghum bicolor* [25], *Nicotiana tabacum* [26], *Saccharum officinarum* [27], *Cucumis sativus* [28], and *Capsicum annuum* [29].

Plants are immobile and thus often subject to various environmental stresses, which results in decreased productivity and growth [30,31]. Plants that are stressed by fungi, bacteria, and insects produce more chitinase, which hydrolyzes chitin and promotes systemic acquired resistance via the hydrolysis of chitin [32]. Chitinase-induced expression is mainly responsible for defense against fungal pathogens [33]. Transgenic maize can be made more resistant to pathogens by overexpressing chitinase [34]. Stress induced by hormones and abiotic factors is also controlled by chitinases [35]. For example, induction of the expression of *BjChi* and *SafChiA* can be triggered by jasmonic acid and mechanical wounding [36,37], and chitinase gene expression can be enhanced by conditions like heavy metal exposure, low temperatures, drought, and high salinity [38,39]. Furthermore, chitinases that are constitutively expressed are associated with various physiological and morphological processes such as seed germination, flowering, aging, and embryogenesis [40,41,42].

Maize (*Zea mays* L.), a key cereal alongside rice and wheat, represents an annual grass [43]. It is economically important in China as a vital food and feed source and as a key industrial raw material [44]. Nonetheless, maize is susceptible to a range of environmental stresses that can adversely impact its productivity and quality. Therefore, identifying endogenous stress-resistance genes, along with breeding stress-resistant varieties, is important for improving maize yield and quality. The approach of analyzing plant gene families can be used to study the evolution and function of plant gene families from a genomics point of view so as to mine the related stress tolerance genes [45,46]. Using high-quality genomic information of maize, many gene families, such as *USP* [47], *CRF* [48], *ACR* [49], *LAC* [50], and *YSL* families [51], have been identified in maize, and related stress tolerance genes have been mined. However, the number of identified maize chitinase gene family members may be incomplete due to the constant updating of maize genome versions and differences between databases [52,53,54]. Previous research on the transcriptional responses of the maize chitinase gene family under various abiotic and biotic stress conditions has been limited, which has restricted the understanding of their biological functions in maize.

This research employed bioinformatics tools to discover chitinase family genes in the maize B73_V5 genome. We investigated their chromosomal positions, structures of genes and proteins, phylogenetic connections, functional enrichment, and collinearity. Furthermore, we leveraged the maize B73_V5 genome to reassess transcriptomic data derived from extensive maize sequencing projects. A study of chitinase gene expression was performed at various developmental stages and under different stress conditions. Our results suggest that genes of the chitinase family contribute to maize growth and development. This research provides insights into how chitinase family members are regulated during stress in maize, and identifies genes that may be useful for breeding maize varieties that are resistant to stress.

## 2. Materials and Methods

### 2.1. Family Membership Identification and Chromosome Localisation

To identify the members of the maize chitinase gene family, an extensive and comprehensive review of the existing literature was conducted. This review was crucial in understanding the current knowledge surrounding the chitinase gene family in maize. Following this literature review, relevant genomic data were gathered, specifically from the B73_V5 genome sequence, mRNA sequences, and protein sequence files, which were obtained from the Phytozome database [55]. This step was essential to ensure that the analysis was based on the most accurate and updated genomic information available for maize.Using the data acquired from these sources, an on-site protein database was established. This custom database served as a foundational resource for the further analysis of maize chitinase proteins. A TAIR database was also used to obtain protein sequences for *A. thaliana* chitinases. [56]. This inclusion allowed for comparative analyses between the chitinase proteins of maize and those of Arabidopsis, enhancing the understanding of the evolutionary relationships between these species.In order to identify sequences that contain structural domains characteristic of chitinase proteins, we utilized HMM model files (PF00187, PF00704, and PF00182) that are associated with the chitinase gene family [57]. For the actual identification of the relevant sequences, we employed the HMMER v3.3.2 web server alongside the local BLASTP program [58]. Candidate genes were identified using a screening threshold of E < 1e^−10^. Sequence information from these candidate proteins was extracted, and redundant sequences were removed. Potential chitinase gene sequences were validated using additional databases, specifically SMART and CDD, to accurately pinpoint members of the chitinase enzyme gene family in maize [59,60]. It was mapped by using the TBtools software (version 2.086) that maize chitinase genes were distributed chromosomally [61], with “Chr N” denoting the specific chromosome locations of the sequences (Please refer to Appendix A for the URLs and access times of all the websites in Section 2).

### 2.2. Structural Characterization of Physicochemical Properties, Genes, and Proteins of the Maize Chitinase Gene Family

The ExPASy platform facilitated an in-depth analysis of the physicochemical characteristics of maize chitinase genes, assessing factors including amino acid profile, grand average hydropathy, isoelectric point, instability index, molecular weight, and aliphatic index [62]. Predictions of subcellular localization for these genes were made using WoLF PSORT [63] and Cell-PLoc [64] software. Domain information for the maize chitinase genes was obtained from the NCBI database [65], while motif analysis was conducted using MEME software, with the number of motifs capped at 12 [66]. Subsequently, TBtools software was employed to delineate exon-intron organizations, domains, and conserved motifs within the maize chitinase gene family.

### 2.3. Phylogenetic Examination of the Maize Chitinase Gene Family

To understand how maize chitinase relates evolutionarily to those in other plants, we selected 24 chitinase proteins from Arabidopsis [23] and 45 from rice [67], drawing on existing research into their gene families. The sequences of chitinase proteins in maize, *Arabidopsis thaliana* and rice were multiplexed in MEGA (version 10) software using default parameters [68], based on the comparison results, 1000 bootstrap replications were used to construct a phylogenetic tree using the neighbor-joining method. Evolutionary trees were visualised and optimised by Evolview v2 [69].

### 2.4. Chitinase Family Gene Collinearity Analysis in Maize

To investigate the collinearity of maize chitinase genes within their species and in comparison to genes from Arabidopsis and rice, TBtools software was utilized. The MCScanX tool in TBtools was used to analyze gene duplication events, and the rates of non-synonymous (Ka) to synonymous (Ks) substitutions were calculated [70]. A Ka/Ks ratio below 1 indicates purifying selection, a ratio of 1 suggests neutral evolution, and a ratio above 1 implies positive selection. The divergence time was approximated using the formula T = Ks/(2λ), with λ assumed to be 6.5 × 10^−9^ [71,72].

### 2.5. Cis-Acting Elements and Functional Analyses of Maize Chitinase Genes

A 2000-base pair promoter region upstream of maize chitinase genes was examined using the PlantCare tool to predict gene functions [73]. Gene ontology (GO) annotations specific to maize were then retrieved from the ShinyGO v0.741 online resource by entering potential candidate proteins or gene identifiers [74]. GO enrichment analysis was conducted with an FDR threshold set at 0.01, and the results were graphically depicted using the Microbiology Letter online visualization tool [75].

### 2.6. Tissue-Specific and Stress-Induced Expression Profiles of the Maize Chitinase Gene Family

Transcriptome sequencing data from various maize tissues under stress were obtained from NCBI. Sequences from the SRA database were converted to Fastq format using the SRA Toolkit. FastQC (version 0.12.1) assessed the data quality [76], and Trimmomatic (version 0.40) was used to filter out low-quality sequences, producing clean datasets [77]. The TBtools software was employed, incorporating Hisat2-Build for indexing, Hisat2-Align for read alignment, Stringtie-Assembly for assembling transcripts, and Stringtie-Quantify for estimating expression levels. The processed and filtered data were aligned against the maize B73_V5 genome to produce FPKM files, facilitating the estimation of gene expression levels. TBtools’ HeatMap tool was then used to perform a differential gene expression analysis on the count data of each gene.

## 3. Results

### 3.1. Basic Information on Members of the Maize Chitinase Gene Family

The chitinase gene family’s Hidden Markov Model (HMM), created from the chitinase protein sequence of *Arabidopsis thaliana* and the Pfam database, was analyzed using BLAST with the genomic data of maize B73_V5 from the Phytozome database. In the complete genome of maize B73_V5, a total of 43 chitinase genes were identified. Subsequent analysis of the amino acid sequence characteristics of the maize chitinase gene family revealed that the coding sequence (CDS) length varied from 285 to 2172 base pairs, corresponding to amino acid counts ranging from 95 to 724 and molecular weights between 10.56 and 77.31 kilodaltons. The calculated isoelectric points of these proteins extended from 4.06 to 10.35. Additionally, the protein composition was found to consist of 44.19% basic proteins and 55.81% acidic proteins. The instability coefficient ranged from 23.99 to 67.31; 18 chitinase proteins were classified as stable, with instability indices above 40, while 25 chitinase proteins were deemed unstable, with instability indices below 40. The lipid index ranged from 41.76 to 95.00. The overall mean hydrophobicity of the 36 chitinase proteins was less than 0, indicating that they are hydrophilic (Appendix A). Subcellular localization predictions indicate that maize chitinase proteins are predominantly situated in chloroplasts (Figure 1) (Appendix A).

### 3.2. Chitinase Genes in Maize Are Distributed across Various Chromosomes

A chromosome map illustrating the distribution of chitinase genes was created by evaluating the genomic locations of the 43 maize chitinase gene family members (Figure 2). Aside from chromosome 9, there are 43 chitinase genes spread unevenly throughout the maize chromosomes. Chromosome 6 contains the most chitinase genes with eight, followed by chromosome 7 with seven, while chromosome 10 has the least with two. The chitinase genes on chromosomes 2, 3, 5, 6, and 7 predominantly occurred in gene clusters.

In gene replication events, 15 gene pairs were identified (9 tandem replications and 6 fragment replications) based on evolutionary relationships and chromosomal localization. The study findings indicate that the maize chitinase gene family has expanded predominantly via tandem duplications, although fragmented duplications have also contributed to this process. This indicates that the main factor behind the growth of the maize chitinase gene family is intra-chromosomal gene duplication. Specifically, three replications were detected for both *ZmChi6* and *ZmChi18*, including both tandem and segmental replications. Indicating that *ZmChi6* and *ZmChi18* are highly active and play a major role in the gene family’s expansion. The phenomenon of gene differentiation resulting from fragment replication is estimated to have taken place between 17.74 million and 53.83 million years ago. In contrast, the timeline for tandem replication is believed to span from 3.95 million to 58.36 million years ago. A thorough analysis of 15 replicated gene pairs demonstrated that the rate of Ka was consistently lower than the rate of Ks. This observation indicates a clear selective bias favoring synonymous substitutions, which is crucial in maintaining functional roles of the genes (Appendix A). These 15 gene pairs have a Ka/Ks ratio under 1 as well, supporting the notion that these genes have experienced substantial purifying selection. This selective pressure has worked to preserve their functional integrity, resulting in minimal divergence over time. The implications of this finding underscore the evolutionary significance of these genes, as they have maintained their vital roles through natural selection, ensuring their proper functionality in the biological context.

### 3.3. Phylogenetic Analysis of Maize Chitinase Genes

Based on the previous studies on chitinase gene families in *Arabidopsis thaliana* and rice, a phylogenetic tree was constructed by combining the classification of GH families in order to deeply investigate the genetic relationship and biological functions of maize chitinase gene families (Figure 3). The chitinase genes were classified into six groups with varying numbers of genes in each group by multiple sequence alignment and phylogenetic analysis.

Subgroup III emerged as the category with the greatest representation of chitinase genes, containing a total of 14 genes. In contrast, subgroup II was found to have 9 genes, while subgroups IV and V were each characterized by the presence of 6 genes. Additionally, subgroups I and VIII each comprised 4 genes. Notably, subpopulation VIII contains only maize chitinase genes and all belong to the GH20 family. Collectively, we identified 26 chitinase genes that displayed direct homology, accounting for approximately 23.21% of the overall gene pool analyzed in this study. In our comparative analysis, we identified thirteen orthologous gene pairs shared between maize and sorghum, which indicates a significant genetic connection between these two species, along with the existence of multiple orthologous genes. Conversely, no orthologs were identified when comparing maize with *Arabidopsis thaliana*, suggesting a divergence in the genetic relationships among these plants. Additionally, our investigation unveiled 26 pairs of paralogous genes within each species examined. Of these, 11 pairs were found in rice, 9 pairs in maize, and 6 pairs in *Arabidopsis thaliana*, highlighting the genetic complexity and diversity within these plant species (Appendix A).

### 3.4. Gene Structure and Protein Motifs of the Maize Chitinase Gene Family

TBtools software was used to group and analyze maize chitinase genes, facilitating the elucidation of their gene structures and protein motifs. These genes were categorized into six unique subfamilies, designated as groups I, II, III, IV, V, and VIII (Figure 4). The clustering of chitinase genes in maize, Arabidopsis, and rice corroborates these observations (Figure 3). Subgroups VI and VII were devoid of any maize chitinase genes.

The conserved sequences among chitinase proteins varied across different subgroups, and proteins within the same subgroup had identically conserved sequences. For example, the 14 ZmChi proteins in subgroup III were found to possess motifs such as Motif (2, 3, 4, 5, 6, and 9).. In contrast, the 13 ZmChi proteins in subgroups I and II contained motifs such as Motif (1, 8, 11, and 12), which were all arranged in the same sequence. The six ZmChi genes in subgroup V were devoid of any motifs, which might imply that the functional diversification observed in maize chitinase genes could be a result of their varied distribution among different evolutionary lineages (Appendix A). Within a subgroup of chitinase genes, similar conserved motifs suggest potential functional similarities. All gene family members had 1 to 15 exons, and some members were devoid of introns. The intron count ranges from 0 to 15, which is quite divergent compared with chitinases in other plants, and this indicates that introns in chitinase genes are not highly conserved across various plant species.

### 3.5. Functional Analysis of Chitinase Family Genes in Maize

Promotor sequence analysis within the maize chitinase gene family uncovered 15 distinct classes of cis-acting regulatory elements, with methyl jasmonate-responsive elements being the most prevalent, comprising 24.6% of the total, as shown in Figure 5. They were closely trailed by elements involved in light signaling, which constituted 20.7%. In addition, a variety of elements linked to hormonal responses—encompassing salicylic acid, auxin, abscisic acid, and gibberellin—were noted, alongside those associated with stress reactions, such as to drought, cold, and hypoxia. Elements implicated in defense and stress, as well as those influencing seed-specific, meristematic, and endospermic gene expression, were also identified.

ShinyGO v0.741 was used for Gene Ontology Enrichment Analysis; functional annotations were obtained for 43 maize chitinases (Figure 6). The enrichment analysis of biological functions indicated that the ZmChi genes were predominantly associated with processes such as carbohydrate metabolism, chitin degradation, and overall catabolism (Figure 6A). Moreover, enrichment analysis of cellular functions revealed that ZmChi genes were specifically associated with the extracellular compartment (Figure 6B). In terms of cellular functions, ZmChi genes exhibited significant enrichment in hydrolase activity, chitinase activity, and carbohydrate derivative binding (Figure 6C).

### 3.6. Colinearity of Maize Chitinase Family Genes

Gene duplication and divergence mechanisms underpin the genesis of novel gene families and functions. To elucidate the gene duplication events in maize relative to other species, both tandem and segmental duplications were characterized to shed light on the evolutionary trajectory of maize chitinase genes. The interspecific collinearity between chitinase genes in maize, *Arabidopsis thaliana*, and rice was examined using TBtools software (Figure 7A). We identified 30 collinear chitinase gene pairs, including one pair between maize and *Arabidopsis thaliana* and 29 pairs between maize and rice. The *ZmChi27* gene was detected in three collinear pairs. Several genes, including *ZmChi1*, *ZmChi6*, *ZmChi18*, *ZmChi20*, *ZmChi25*, *ZmChi29*, *ZmChi40*, and *ZmChi43* were each detected in two collinear pairs. *ZmChi3*, *ZmChi9*, *ZmChi13*, *ZmChi22*, *ZmChi26*, *ZmChi32*, *ZmChi35*, *ZmChi36*, *ZmChi39*, *ZmChi41*, and *ZmChi42* were each detected in one collinear pair. The remaining maize chitinase genes did not display any collinear relationships with the species examined.

Furthermore, the Ka/Ks ratio, which compares the frequency of amino acid-altering to silent mutations, serves as a crucial indicator of the types of selection pressures acting on duplicated genes. This metric is instrumental in deciphering the evolutionary pressures and selection acting upon protein-coding sequences. This study calculated Ks, Ka, and Ka/Ks for each gene pair and found that tandemly duplicated maize chitinase genes had significantly lower mean values than segmentally duplicated genes. Aside from maize homologous gene pairs and rice homologous gene pairs, which had Ks values over 1, the Ka and Ka/Ks values for maize homologous gene pairs, as well as Arabidopsis and rice homologous gene pairs, were all less than 1 (Figure 7B). Maize chitinase gene duplication events were estimated to have occurred approximately 24.52 and 40.94 million years ago (Mya), and maize and rice diverged at 45.38 Mya.

### 3.7. Tissue-Specific Expression Profiles of Maize Chitinase Genes

Employing transcriptome sequencing data across diverse maize tissues (SRP010680) [78], an expression heat map of maize chitinase genes was generated using maize B73_V5 genome data (Figure 8). Only a few maize chitinase family genes showed high expression levels prior to 10DAP_Whole seeds and low expression levels after 10DAP_Whole seeds. *ZmChi1* demonstrated elevated transcript abundance before the 10-day post-pollination mark in entire seeds. *ZmChi43* and other genes were highly expressed after 10DAP_Whole seeds. *ZmChi35*, *ZmChi22*, and *ZmChi29* were highly expressed across all tissues, suggesting that they play a role in various physiological processes during maize development. Conversely, the majority of chitinase family genes exhibited low expression levels across maize tissues. Ten genes, including *ZmChi4*, *ZmChi8*, *ZmChi10*, *ZmChi19*, *ZmChi21*, *ZmChi25*, *ZmChi27*, *ZmChi36*, *ZmChi37*, and *ZmChi40*, were not expressed in most maize tissues. *ZmChi6* were highly expressed only in 24H_Germinating Seed and were not expressed in other tissues, *ZmChi28* were highly expressed in 24H_Germinating Seed and expressed slightly in Stem and SAM and almost undetectable in root. These distinct expression patterns suggest unique roles for maize chitinase genes in different tissues.

### 3.8. Analysis of Maize Chitinase Gene Family Expression Profiles under Abiotic Stress

Leveraging publicly accessible transcriptome sequencing data for maize exposed to diverse abiotic stressors, such as temperature extremes (PRJNA645274) [79], waterlogging stress (PRJNA606824) [80], drought stress (PRJNA576545) [81], salt stress (PRJNA414300) [82], and light fluctuation (PRJNA408209) [83], a transcriptome analysis was performed, utilizing the maize B73_V5 genome as a reference. Heatmap visualizations were subsequently created to depict the transcriptional responses of the maize chitinase gene family under various stress conditions (Figure 9). The study disclosed that these genes’ expression was elevated under waterlogged conditions, yet reduced under high and low-temperature stress. Notably, *ZmChi15*, *ZmChi26*, and *ZmChi31* exhibited robust expression across a range of stressors. Genes that exhibited differential expression were found to be significantly downregulated under extremes of temperature, waterlogging, and drought stress when compared to the non-stressed control group. Conversely, under salt stress and variable light conditions, gene expression was markedly upregulated compared to the control.

Compared to the control, the expression levels of 10 specific chitinase genes (*ZmChi5*, *ZmChi7*, *ZmChi13*, *ZmChi15*, *ZmChi17*, *ZmChi20*, *ZmChi26*, *ZmChi28*, *ZmChi31*, and *ZmChi39*) decreased under temperature stress conditions (Figure 9A). In contrast, the expression of three chitinase genes (*ZmChi3*, *ZmChi4*, and *ZmChi9*) was significantly altered under different temperature stress conditions. Under low-temperature stress, *ZmChi3* and *ZmChi4* expression was significantly up-regulated, whereas *ZmChi9* expression was significantly up-regulated. Under waterlogging stress (Figure 9B), 10 chitinase genes (*ZmChi7*, *ZmChi13*, *ZmChi14*, *ZmChi20*, *ZmChi22*, *ZmChi28*, *ZmChi30*, *ZmChi31*, *ZmChi35*, and *ZmChi40*) were highly expressed. A significant increase in chitinase gene expression was seen under waterlogging stress (*ZmChi14*, *ZmChi20*, *ZmChi28*, *ZmChi30*, and *ZmChi31*). The minimum number of genes with significant expression occurred at the 2-h mark of the flooding treatment, with the peak expression observed after 4 h of treatment. Under drought stress (Figure 9C), three chitinase genes (*ZmChi1*, *ZmChi4*, and *ZmChi13*) were highly expressed, and the expression of drought-tolerant hybrids and drought-intolerant hybrids was significantly down-regulated compared with the control. The expression of six chitinase genes (*ZmChi7*, *ZmChi13*, *ZmChi15*, *ZmChi26*, *ZmChi31*, and *ZmChi43*) was highly expressed under salt stress conditions (Figure 9D). Under salt stress, the expression levels of 12 chitinase genes (*ZmChi7*, *ZmChi8*, *ZmChi13*, *ZmChi14*, *ZmChi15*, *ZmChi22*, *ZmChi26*, *ZmChi28*, *ZmChi31*, *ZmChi37*, *ZmChi39*, and *ZmChi43*) were markedly elevated in salt-tolerant and both salt-sensitive maize varieties compared to control conditions. Conversely, the expression of two chitinase genes (*ZmChi5* and *ZmChi20*) was significantly down-regulated exclusively in salt-sensitive varieties. Under light fluctuations (Figure 9E), four chitinase genes (*ZmChi13*, *ZmChi15*, *ZmChi26*, and *ZmChi31*) were highly expressed. The expression patterns of five chitinase genes (*ZmChi5*, *ZmChi27*, *ZmChi29*, *ZmChi31*, and *ZmChi41*) were up-regulated to similar degrees under light-fluctuating conditions compared with the control. The genes *ZmChi5*, *ZmChi31*, and *ZmChi41* exhibited significant upregulation in expression, whereas *ZmChi27* and *ZmChi29* showed substantial downregulation.

### 3.9. Analysis of Maize Chitinase Gene Family Expression Profiles under Biotic Stress

This research is based on transcriptome sequencing data from various biotic stress conditions—such as maize smut (*Sphacelotheca reiliana*) (PRJNA673988) [84], stem rot (*Fusarium graminearum* Schwabe) (PRJNA721468) [85], leaf spot (*Mycosphaerella maydis*) (PRJNA436207) [86], beet armyworm stress (*Spodoptera exigua*) (PRJNA625224) [87], asian corn borer(*Ostrinia furnacalis*) (PRJNA772910) [88], and aphid (*Rhopalosiphum maidis*) (PRJNA295410) [89]—were re-examined using the maize B73_V5 genome as a reference. A heatmap was constructed to depict the transcriptional responses of maize chitinase genes to diverse biotic challenges (Figure 10). Interestingly, smut stress significantly increased the expression of maize chitinase genes, while aphid infestations had relatively muted responses. *ZmChi1*, *ZmChi7*, *ZmChi13*, *ZmChi26*, and *ZmChi31* showed strong expression under several biotic stresses, with significant upregulation after exposure to smut, stem rot, Asian corn borer, and aphids compared to controls. However, their expression remained unchanged under leaf spot and beet armyworm stress.

When exposed to smut stress (Figure 10A), the levels of expression of seven maize chitinases (*ZmChi1*, *ZmChi13*, *ZmChi15*, *ZmChi26*, *ZmChi28*, *ZmChi31*, and *ZmChi35*) were elevated. No differences in the expression of seven maize chitinase genes (*ZmChi2*, *ZmChi12*, *ZmChi21*, *ZmChi23*, *ZmChi36*, *ZmChi38*, and *ZmChi40*) under stress and control conditions were observed. The expression of the remaining 36 members was significantly altered; the expression of nine members (*ZmChi1*, *ZmChi3*, *ZmChi7*, *ZmChi13*, *ZmChi15*, *ZmChi20*, *ZmChi26*, *ZmChi28*, and *ZmChi31*) was up-regulated in six maize strains. Following exposure to stem rot (Figure 10B), the expression levels of five maize chitinases (*ZmChi7*, *ZmChi13*, *ZmChi15*, *ZmChi26*, and *ZmChi31*) increased. A greater number of differentially expressed genes were identified between resistant and susceptible plants than in the control group, with the latter being subjected to leaf spot stress for a period of 7 days. Additionally, an escalation in gene expression was noted in correlation with prolonged exposure to stem rot stress. The expression of six maize chitinases (*ZmChi1*, *ZmChi2*, *ZmChi4*, *ZmChi13*, *ZmChi20*, and *ZmChi26*) was higher under leaf spot disease stress (Figure 10C), and all six chitinase genes showed significant expression when subjected to leaf spot disease stress; in addition, more significantly expressed genes were found in resistant than in susceptible compared to the control material, and the susceptible plants were under leaf spot disease stress under the treated for 7 days had the most significantly expressed differential genes.

Under beet armyworm stress (Figure 10D), five maize chitinases (*ZmChi7*, *ZmChi13*, *ZmChi15*, *ZmChi26*, and *ZmChi31*) were highly expressed. The expression of 12 maize chitinase genes (*ZmChi1*, *ZmChi5*, *ZmChi7*, *ZmChi13*, *ZmChi15*, *ZmChi17*, *ZmChi20*, *ZmChi26*, *ZmChi28*, *ZmChi29*, *ZmChi31*, and *ZmChi35*) was significantly up-regulated, and the expression levels of *ZmChi27* were down-regulated. Under Asian corn borer stress (Figure 10E), 10 maize chitinases (*ZmChi1*, *ZmChi7*, *ZmChi13*, *ZmChi18*, *ZmChi25*, *ZmChi26*, *ZmChi28*, *ZmChi29*, *ZmChi31*, and *ZmChi35*) were expressed at higher levels; and, in comparison with control material, 18 maize chitinase genes (*ZmChi3*, *ZmChi6*, *ZmChi8*, *ZmChi10*, *ZmChi11*, *ZmChi12*, *ZmChi16*, *ZmChi21*, *ZmChi23*, *ZmChi24*, *ZmChi27*, *ZmChi32*, *ZmChi33*, *ZmChi34*, *ZmChi38*, *ZmChi40*, *ZmChi42* and *ZmChi43*) were not significantly expressed, *ZmChi1*, *ZmChi2*, *ZmChi4*, and *ZmChi36* were significantly down-regulated, Conversely, the remaining 21 chitinase genes in maize demonstrated substantial upregulation. Under aphid stress (Figure 10F), the expression levels of three maize chitinases (*ZmChi3*, *ZmChi13*, and *ZmChi26*) were up-regulated. Under aphid stress, 10 particular maize chitinase genes (*ZmChi1*, *ZmChi5*, *ZmChi13*, *ZmChi15*, *ZmChi17*, *ZmChi20*, *ZmChi26*, *ZmChi30*, *ZmChi31*, and *ZmChi35*) displayed notable upregulation relative to control conditions.

### 3.10. Biological and Abiotic Stress Regulation of Maize Chitinase Genes

Investigating the expression of maize chitinase genes under a range of abiotic and biotic stresses revealed numerous genes with differential expression, as depicted in the heatmap (Figure 11). These genes showed diverse expression patterns under various abiotic stresses and were predominantly upregulated in the face of biotic challenges. The most pronounced changes in expression were associated with smut and stem rot diseases, whereas the subtlest responses were seen with beet armyworm and aphid attacks. Of particular note, a significant upregulation of gene expression was detected in response to smut stress. Conversely, under abiotic stresses such as temperature fluctuations and waterlogging, gene expression was down-regulated. Furthermore, the most significant differential gene expression was observed in response to leaf spot infection.

This study’s findings indicated that the 41 identified maize chitinase genes are vital in stress response mechanisms. Among these genes, the expression levels of *ZmChi5* and *ZmChi31* were particularly noteworthy, as they displayed significant sensitivity to all 11 stress conditions that were investigated. In particular, the expression of *ZmChi31* was considerably down-regulated when exposed to temperature stress, contrasting with its up-regulation under the other 10 stress conditions. This distinct expression profile implies that *ZmChi31* could be instrumental in the plant’s stress response. Consequently, *ZmChi31* might be a potential candidate for future studies on stress response mechanisms in maize. The expression of five chitinase genes in maize (*ZmChi13*, *ZmChi15*, *ZmChi20*, *ZmChi26*, and *ZmChi30*) significantly responded to 10 stresses. Transcription of *ZmChi13*, *ZmChi15*, and *ZmChi26* was significantly enhanced under a variety of biotic stress conditions, yet was suppressed under abiotic stress, and increased under salt stress. The expression of *ZmChi23* significantly responded only to smut, and the expression of *ZmChi12* and *ZmChi38* did not respond significantly to any stress conditions.

## 4. Discussion

The swift progress in genomic sequencing has catalyzed a paradigm shift in biological research, notably within plant genomics [90,91,92]. The identification of gene families is important for elucidating biodiversity and evolutionary processes, which enhances our understanding of genetic variation across different species [93]. Nevertheless, sequence diversity among gene family members poses a significant challenge to gene identification; traditional sequence homology-based methodologies are often not effective for identifying all related genes, especially those exhibiting low sequence similarity. Chitinases, enzymes that catalyze the breakdown of chitin—a polysaccharide crucial for fungal cell walls—are pivotal in plant defense, particularly against fungal pathogens [94]. Additionally, these genes are significant in plant stress tolerance, development, and growth [94,95]. Chitinase research has been conducted extensively across a range of plant species, including *Triticum aestivum* [22], *Arabidopsis thaliana* [23], *Glycine max* [24], *Sorghum bicolor* [25], *Nicotiana tabacum* [26], *Saccharum officinarum* [27], *Cucumis sativus* [28], and *Capsicum annuum* [29].

Although the identification of maize chitinase gene families has been reported in several studies, the number of maize chitinase gene family members identified in previous studies (39) was lower than that identified in the present study (43) due to differences in genome versioning and annotation updates, gene prediction, and annotation methods, suggesting that the genome-wide identification of maize chitinase family genes in previous studies was incomplete [54]. In this study, a comprehensive gene family analysis of the maize chitinase gene family was performed, especially covariance analysis and functional prediction, which helped to understand the evolution and functional differentiation of these genes. In terms of gene evolution, the present study revealed that the duplication events of tandem and segmental duplicated genes of maize chitinase genes occurred at about 24.52 and 40.94 Mya (million years ago), and maize diverged from rice at 45.38 Mya. In terms of functional prediction, the maize chitinase genes were found to have important roles in carbohydrate metabolism, chitinolytic metabolism and catabolism.

Chitinase gene expression is known to be sensitive to harsh environmental conditions. For instance, chitinase genes in class IV and VII are crucial in conferring wheat resistance against *Fusarium graminearum* infections [96]. Davis et al. found that inoculating pineapple with Fusarium induces the production of class IV and I chitinases [97]. In 2020, Liu et al. thoroughly analyzed the class II chitinase gene *LcCHI2* in *Leymus chinensis*. Their study showed that the elevated expression of *LcCHI2* resulted in enhanced chitinase activity, thereby substantially boosting the resistance of transgenic tobacco and maize to a range of pathogens [98]. This finding underscores the potential of chitinase genes as a means of enhancing plant defense mechanisms, suggesting that manipulation of these genes could be a viable strategy for increasing crop resilience. Furthermore, research conducted by Hawkins et al. illuminated the potential protective functions of maize chitinase genes against aflatoxins, along with the reduction of their accumulation within the plant [53]. This highlights the multifaceted roles that chitinase genes can play not only in combating pathogens but also in mitigating the effects of toxic compounds that can adversely affect crop yield and quality [99]. Additional studies provide further evidence of the involvement of chitinase genes in protecting plants from pest infestations. For instance, the transformation of tobacco plants with the *SmchiC* chitinase gene, which is derived from the bacterium *Serratia marcescens*, has resulted in the creation of transgenic plants that show improved resistance to both the fungal pathogen *Botrytis cinerea* and the insect pest *Spodoptera frugiperda* [100]. These advancements suggest a promising avenue for agricultural biotechnology in developing crops that are more resilient to biotic stressors, thereby enhancing food security and sustainability.

Additionally, studies suggest that chitinases are involved in plant reactions to both abiotic and biotic stressors [8]. Jasmonic acid and injury can trigger the activation of the expression of *BjCHI* and SafChia genes, whereas environmental factors such as excessive salt, cold temperatures, lack of water, and toxic metals can enhance the synthesis of chitinase genes in plants [4]. Transgenic tobacco plants that express *Trichoderma harzianum* chitinases *CHIT33* and *CHIT42* exhibit enhanced tolerance to heavy metals, salt stress, and pathogen resistance [101,102]. The induced expression of maize chitinase may be related to signalling pathways in plants. For example, heightened chitinase activity could be associated with the production of signaling molecules, such as ROS, within plant cells. These molecules initiate a broad spectrum of defense mechanisms when plants experience stress [31]. Further, chitinase enzymes play an important role in various physiological and developmental processes in plants, including embryogenesis, flowering, aging, and seed germination [103,104,105].

The chitinase gene family plays a key role in the regulation of plant growth and development and in response to biotic and abiotic stresses [36,105]. Previous studies have focused on the function of chitinases from the perspective of a single specific stress [52,53,54]. In the present study, we used a large-scale transcriptome dataset (304 transcriptome data sets covering 11 different types of stress conditions) to systematically analyse the expression changes of maize chitinase genes under diverse tissues and adverse conditions. Through this comprehensive analysis, we revealed the potential functions and regulatory modes of the maize chitinase gene family in response to a variety of adversities, and provided new perspectives for an in-depth understanding of the complex mechanisms of their roles in plant adversity response. The results suggest that chitinases exert a complex influence on plant growth and development, as well as on their responses to diverse stress conditions. Specifically, maize chitinase genes are integral to the plant’s defense strategies against a wide range of biotic and abiotic stresses. The expression levels of these genes exhibit significant variation when subjected to different abiotic stressors. Importantly, there is a marked upregulation of these genes under biotic stress, with the most elevated expression levels occurring in response to smut and stem rot infections. In contrast, the minimal levels of expression are observed during herbivory from beet armyworms and aphids. Especially under conditions of smut stress, a significant up-regulation of maize chitinase gene expression is noted. In contrast, gene expression is predominantly down-regulated under conditions of temperature stress and waterlogging. *ZmChi5* and *ZmChi31* significantly responded to 11 different types of stress; *ZmChi31* was the only gene that was down-regulated under temperature stress, and it was differentially expressed under leaf spot stress. According to these findings, *ZmChi5* and *ZmChi31* play an important role in stress responses. These genes exhibited heightened expression in reaction to nine additional stressors, suggesting their participation in stress response mechanisms and rendering them prospective subjects for deeper investigation. The expression of five maize chitinases (*ZmChi13*, *ZmChi15*, *ZmChi20*, *ZmChi26*, and *ZmChi30*) significantly increased in response to 10 stress conditions.Notably, three particular maize chitinase genes—*ZmChi13*, *ZmChi15*, and *ZmChi26*—demonstrated a significant upregulation under each of the six biotic stress scenarios. Conversely, their expression was substantially reduced under abiotic temperature stress and notably increased under salt stress conditions. The expression of *ZmChi23* significantly responded to smut, whereas *ZmChi12* and *ZmChi38* were not significantly expressed under any of the stress conditions tested. The transcriptional responses of *ZmChi5* and *ZmChi9* were observed to fluctuate across assorted abiotic and biotic stressors. These genes exhibited distinctive expression profiles, with the most significant variations appearing in their expression levels. Except for *ZmChi12* and *ZmChi38*, the rest of the 41 maize chitinase genes demonstrated substantial stress-responsiveness. Specifically, the expression of *ZmChi21* and ZmChi40 varied under various abiotic stress conditions, and *ZmChi23* and *ZmChi32* displayed significant variation in expression exclusively under different biotic stress conditions. Conversely, *ZmChi11* and *ZmChi24* showed markedly different expression in response to abiotic and biotic stressors, respectively. Maize chitinase genes displayed varied transcriptional responses to different stress conditions. Understanding these distinct profiles is crucial for enhancing our comprehension of chitinase gene functions in maize. This study identifies key chitinase genes in various maize tissues and under different stress types, summarizing their unique expression patterns vital for understanding their roles in maize (Appendix A).

The results suggest that chitinase genes from various plants share analogous functions when facing biotic and abiotic stresses. In maize, *ZmChi31* is predicted to localize within mitochondria, classified under subgroup III of the GH18 family and features motifs characteristic of this family. When examined for functionality, it was found to possess cis-acting elements responsive to abscisic acid, defense mechanisms, stress tolerance, drought, and hypoxia, as well as involvement in enhanced carbohydrate metabolism, chitinolytic activity, and catabolism. Utilizing publicly accessible maize transcriptome sequencing data and genomic information from maize B73_V5, our analysis revealed that the *ZmChi31* gene exhibited significant responses to all 11 stress conditions; its expression decreased under temperature stress, fluctuated in response to leaf spot stress, and was significantly elevated with the other nine stress types, which included waterlogging, drought, salinity, light variations, smut, stem rot, beet armyworm, Asian corn borer, and aphid stress. Additionally, these findings will facilitate future research aimed at uncovering the molecular mechanisms that drive maize’s adaptability to stress conditions. This study’s findings underscore the significance of further investigating chitinase genes, particularly their involvement in responses to abiotic and biotic stress.

## 5. Conclusions

According to this study, we catalogued 43 chitinase genes in maize and conducted an exhaustive analysis of their genomic features, encompassing physicochemical properties, chromosomal positions, structural attributes, phylogenetic ties, functional classifications, and collinearity patterns. By re-evaluating published transcriptome data against the B73_V5 maize genome, we profiled the expression of these chitinase genes across various tissues and under diverse stress conditions. The results highlighted a high level of expression diversity, which is believed to be crucial for maize growth and development. Particularly, *ZmChi31* exhibited significant expression changes in response to multiple abiotic and biotic stressors. This investigation deepens our understanding of the functional roles of maize chitinase genes and may facilitate the selection of genes to enhance maize’s resilience to stress.

## Figures and Tables

**Figure 1 genes-15-01327-f001:**
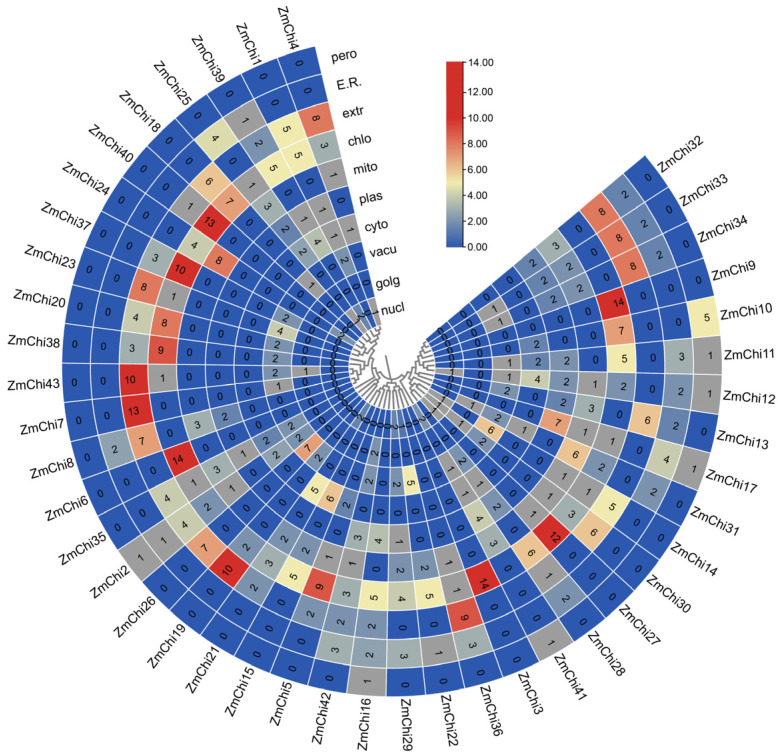
A heat map depicting the subcellular localization of all ZmChi genes within plant cells, including the plasmid, endoplasmic reticulum, Golgi apparatus, vesicles, mitochondria, chloroplasts, nucleus, cytoplasm, ectoplasm, and peroxisomes. Blue indicates the lack of the gene in that area, gray indicates a small functional presence, and red indicates the highest functional importance of the gene in that specific region. Note: nucl: nucleus; golg: Golgi; vacu: vesicle; cyto: cytoplasm; plas: plasmid; mito: mitochondrion; chlo: chloroplast; extr: outer matrix; E.R.: endoplasmic reticulum; pero: peroxisome.

**Figure 2 genes-15-01327-f002:**
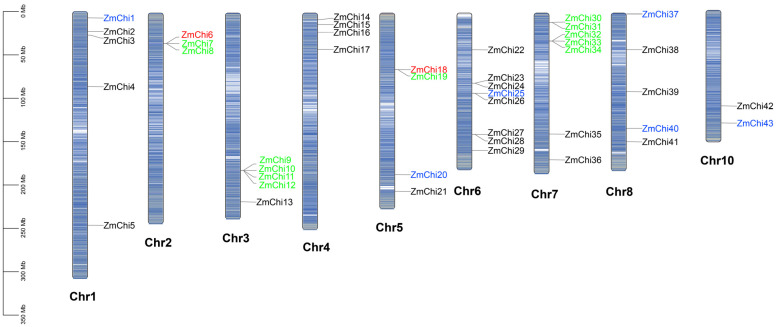
Chromosome distribution of chitinase genes in maize. Note: Genes are color-coded based on duplication type: green for tandem duplicates, blue for segmental duplicates, and red for those that are both tandem and segmental duplicates.

**Figure 3 genes-15-01327-f003:**
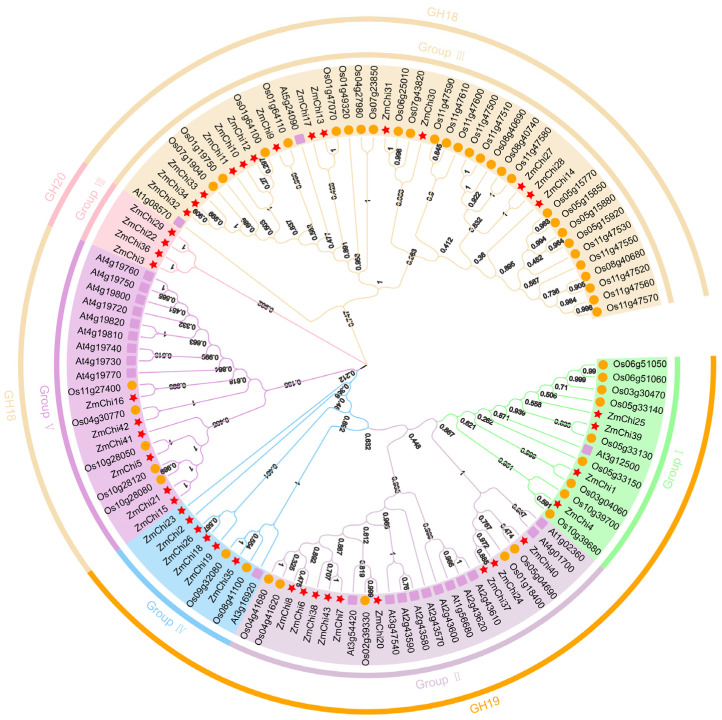
Phylogenetic analysis of rice, maize, and Arabidopsis chitinase proteins. GH18 comprises family members from groups III and V; GH19 encompasses family members from groups I, II, and IV; and GH20 consists of family members from group VIII.

**Figure 4 genes-15-01327-f004:**
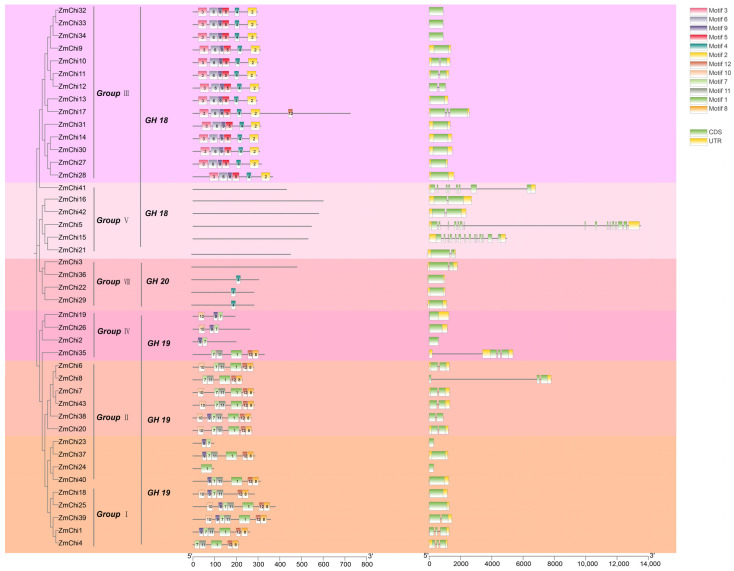
A visual representation showing the preserved pattern of protein motifs and the arrangement of exons and introns in the ZmChi genes of maize. GH18 comprises family members from groups III and V; GH19 encompasses family members from groups I, II, and IV; and GH20 consists of family members from group VIII.

**Figure 5 genes-15-01327-f005:**
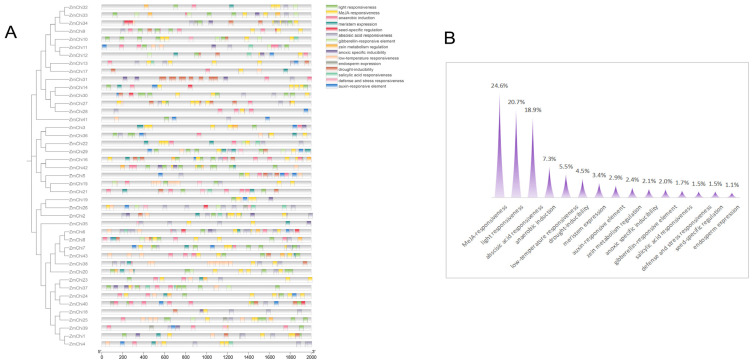
Investigation of cis-regulatory elements within the promoter regions of the maize chitinase gene family: (**A**) Examination of the prevalence of diverse cis-regulatory elements across these promoters. (**B**) Assessment of the frequency of various cis-regulatory element types within these promoter sequences.

**Figure 6 genes-15-01327-f006:**
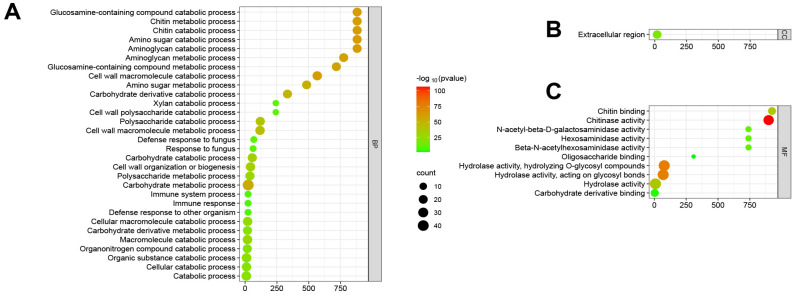
Fold enrichment diagram highlighting the shared functions of ZmChi genes: (**A**) GO terms associated with biological processes; (**B**) GO terms associated with cellular components; (**C**) GO terms associated with molecular functions. Note: Red dot plots indicate a greater number of genes participating in each respective process, and blue dots of smaller size indicate fewer genes.

**Figure 7 genes-15-01327-f007:**
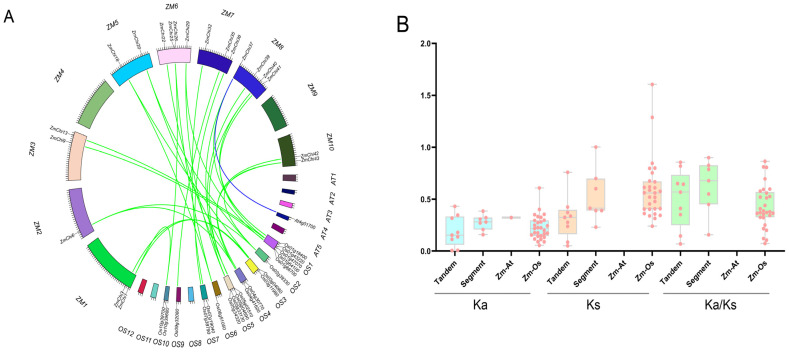
Duplication and synteny of maize chitinase genes. (**A**) The diagram illustrates the homologous relationships of maize chitinase genes with counterparts in *Arabidopsis thaliana* and rice, where green lines denote homologous gene pairs between maize and rice, and blue lines represent those between maize and *Arabidopsis thaliana*. (**B**) The horizontal axis represents tandem repeats, segmental repeats, and maize repeats with *Arabidopsis thaliana* (Zm-At) and rice (Zm-Os). The vertical axis denotes the corresponding numerical values.

**Figure 8 genes-15-01327-f008:**
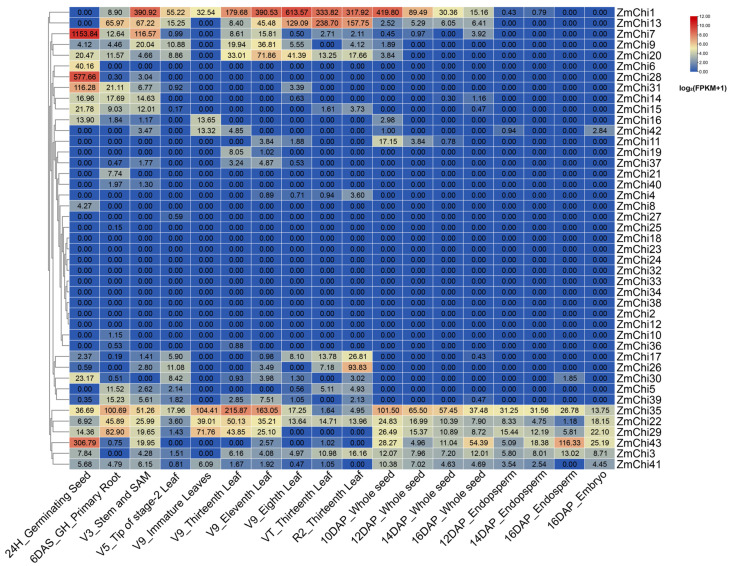
A heatmap shows the heterogeneity in expression patterns of chitinase genes across maize tissues. 24H_Germinating Seed: Seed 24 h after germination begins; 6DAS_GH_Primary Root: Root at 6 days after seed germination; V3_Stem and SAM: Stem and Shoot Apical Meristem (SAM) at the V3 corn growth stage; V5_Tip of stage-2 Leaf: Tip of the stage-2 leaf at the V5 corn growth stage; V9_Immature Leaves: Immature leaves at the V9 corn growth stage; V9_Thirteenth Leaf: Thirteenth leaf at the V9 corn growth stage; V9_Eleventh Leaf: Eleventh leaf at the V9 corn growth stage; V9_Eighth Leaf: Eighth leaf at the V9 corn growth stage; VT_Thirteenth Leaf: Thirteenth leaf at the Tasseling (VT) corn growth stage; R2_Thirteenth Leaf: Thirteenth leaf at the R2 reproductive stage of corn; 10DAP_Whole seed: Whole seed at 10 days after pollination; 12DAP_Whole seed: Whole seed at 12 days after pollination; 14DAP_Whole seed: Whole seed at 14 days after pollination; 16DAP_Whole seed: Whole seed at 16 days after pollination; 12DAP_Endopsperm: Endosperm at 12 days after pollination; 14DAP_Endopsperm: Endosperm at 14 days after pollination; 16DAP_Endopsperm: Endosperm at 16 days after pollination; 16DAP_Embryo: Embryo at 16 days after pollination. Note: The boxed data represent raw FPKM values.

**Figure 9 genes-15-01327-f009:**
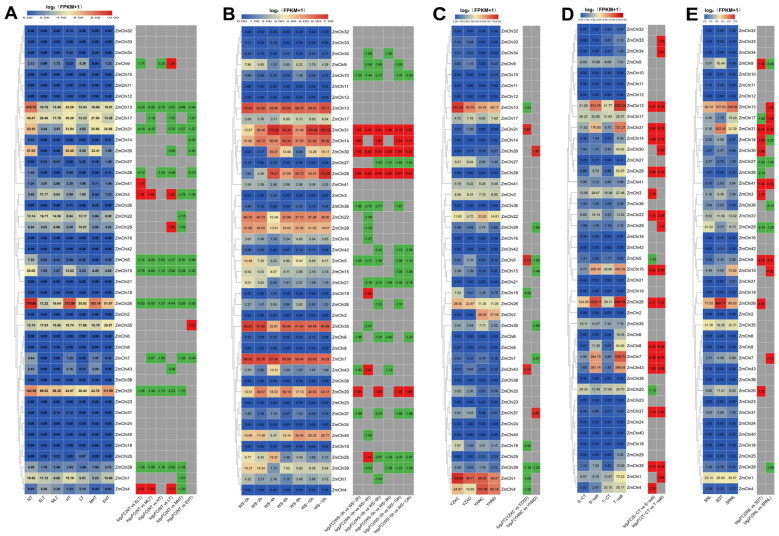
A heatmap displaying maize chitinase gene expression under different abiotic stress conditions. (**A**) The expression patterns of maize chitinase genes were studied under various temperature stress conditions, including NT: normal (25 °C), ELT: extremely low (4 °C), MLT: medium-low (10 °C), HT: high (37 °C), LT: low (16 °C), MHT: medium-high (42 °C), and EHT: extremely high (48 °C) temperatures. (**B**) We analyzed the temporal expression patterns of maize chitinase genes under flooding stress at various intervals: 0 h (ws-0 h), 2 h (ws-2 h), 4 h (ws-4 h), 6 h (ws-6 h), 8 h (ws-8 h), 10 h (ws-10 h), and 12 h (ws-12 h). (**C**) Analysis of the expression of maize chitinase genes under drought stress conditions revealed patterns using different sample types: YZXC for moisture-treated samples from the drought-intolerant hybrid ZX978, YZXD for drought-treated samples from the drought-intolerant hybrid ZX978, YHNC for moisture-treated samples from the drought-tolerant hybrid ND476, and YHND for drought-treated samples from the drought-tolerant hybrid ND476. (**D**) The response of maize chitinase genes to salt stress was investigated by subjecting the salt-tolerant inbred line L87 (T-salt) and the salt-sensitive inbred line L29 (S-salt) to a 220 mM NaCl treatment. Concurrently, control groups T-CT and S-CT for L87 and L29, respectively, were treated with 0 mM NaCl. (**E**) Under varying light conditions, the maize chitinase gene family was evaluated. Here, SNL represented seedlings under standard light conditions; SDT indicated seedlings subjected to four days of darkness; and SRNL denoted seedlings that were first treated with four days of darkness followed by four days of normal light. Note: Each chart depicts the original FPKM value in the left box and the value of log2(multiple changes) in the right box, grey (no significant differences), highlighted in red (up-regulated), and green (down-regulated), |FC| ≥ 1.

**Figure 10 genes-15-01327-f010:**
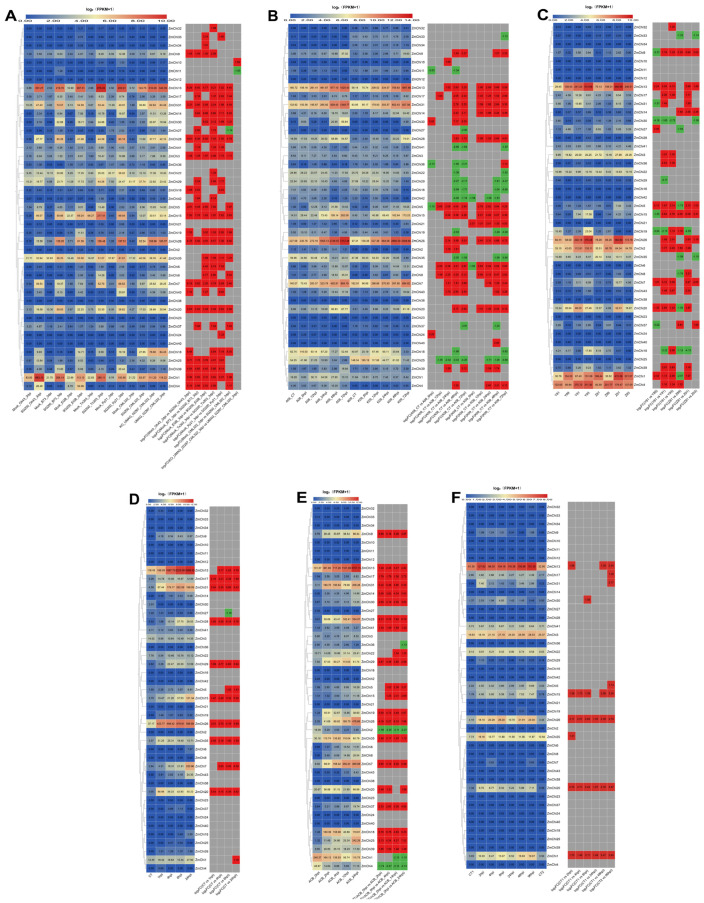
A heatmap displaying maize chitinase gene expression under different biotic stress conditions. (**A**) Gene expression profiles of maize chitinase genes when subjected to black tassel disease stress. The experimental setup included a control treatment (uninfected plants) and treatments with the strains SG 200 and UMAG_02297, which are live nutrient fungi responsible for causing black tassel disease in maize. Additionally, the knockout mutant strain, KO_UMAG_02297, was included. Data were taken 3 days after infection (3 dpi) on six different types of maize: B73, CML 322, EGB, Ky 21, Oh 43, and Tx 303. (**B**) The expression patterns of maize chitinase genes under stem rot stress. Two maize varieties, A08 (resistant) and K09 (susceptible), were infested with *Fusarium graminearum*. The control treatment (CT) consisted of uninfected plants. Plants with pests were examined at different intervals including 6 h after infestation (hpi), 12 hpi, 24 hpi, 48 hpi, and 72 hpi. (**C**) Furthermore, the expression patterns of maize chitinase genes in response to maize gray spot disease stress were analyzed in two different varieties, Y (‘Yayu 889’) and Z (‘Zhenghong 532’); Y and Z are resistant and susceptible to gray spot disease, respectively. Gray spot disease in these plants was evaluated at 81, 89, 91, and 93 days post-infection (dpi). (**D**) The maize chitinase gene family’s response to sugarbeet nightshade moth and Asian corn borer infestation was analyzed. The control group (CT) consists of healthy plants, with 1 hpi, 4 hpi, 6 hpi, and 24 hpi indicating 1, 4, 6, and 24 h post-infestation, respectively. (**E**) The temporal expression of maize chitinase genes following Asian corn borer attack, with time points at 0 hpi, 4 hpi, 12 hpi, and 24 hpi corresponding to 0, 4, 12, and 24 h post-infestation intervals. (**F**) Maize chitinase gene family expression patterns during aphid stress. CT-0 h vs. CT-96 h correspond to 0 h vs. 96 h control treatments (uninfected plants); Aphid-2 h, Aphid-4 h, Aphid-8 h, Aphid-24 h, Aphid-48 h, and Aphid-96 h correspond to 2 h, 4 h, 8 h, 24 h, 48 h, and 96 h after aphid infestation, respectively. Note: Each chart depicts the original FPKM value in the left box and the value of log2(multiple changes) in the right box, highlighted in red (up-regulated) or green (down-regulated), |FC| ≥ 1.

**Figure 11 genes-15-01327-f011:**
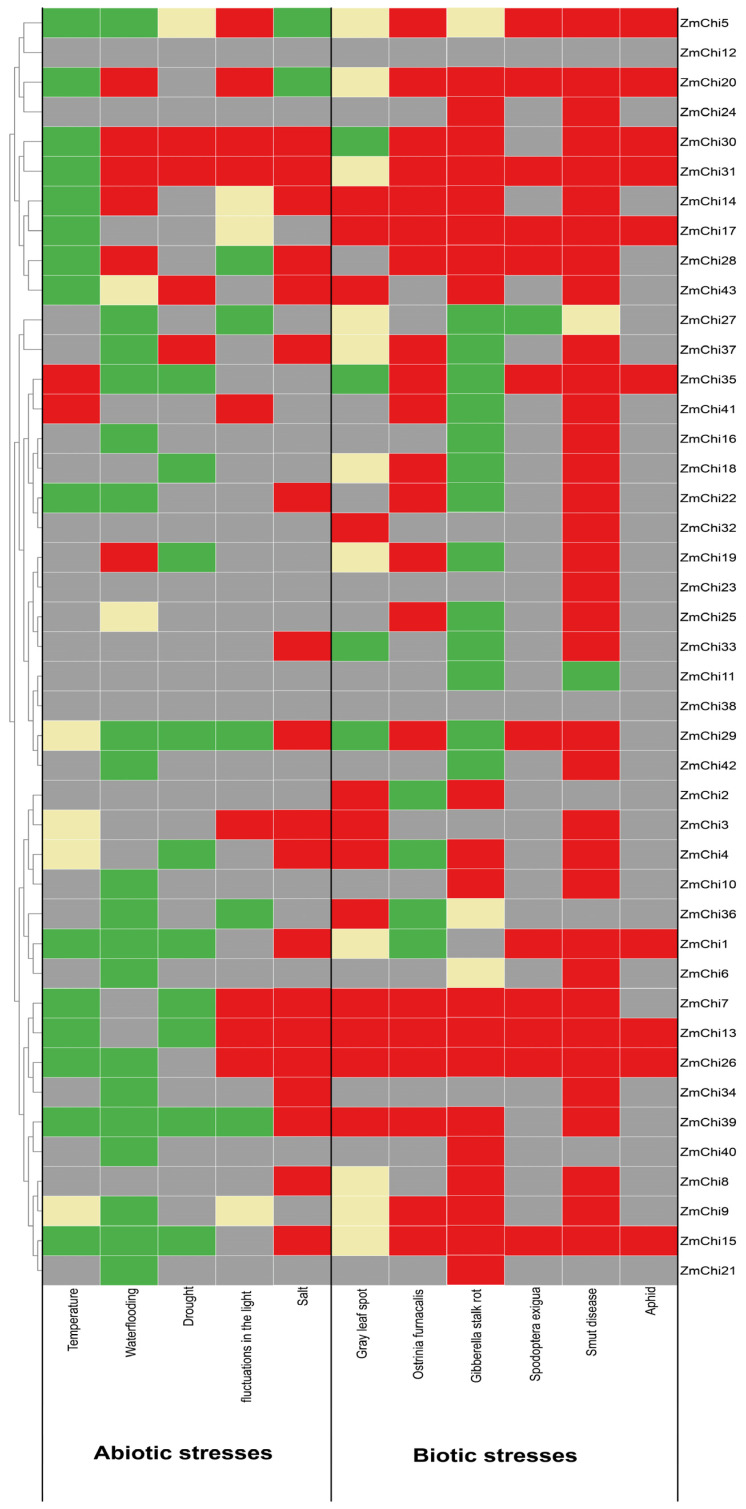
Heatmap depicting the different expression levels of maize chitinase genes in response to abiotic and biotic stresses. Here, gray denotes unchanged expression, red signifies upregulation, green indicates downregulation, and yellow represents a mix of both upregulation and downregulation.

## Data Availability

Data are contained within the article.

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
