# Peer review of "Genome-Wide Identification of the Maize Chitinase Gene Family and Analysis of Its Response to Biotic and Abiotic Stresses"

_genes, 2024, doi:10.3390/genes15101327_

Round 1

Reviewer 1 Report

Comments and Suggestions for Authors

Genome-wide identification of the maize chitinase gene family and analysis of its response to biotic and abiotic stresses

Suggestions for Authors:

The manuscript genes-3251656 presents the study of the gene family of chitinases in maize through a genome-wide analysis, they found tandem duplications and dae the chitinase gene divergence to about 31.16 MYA, they conduct an in silico expression analysis of the 43 genes finding its diffrenial plant tissue expression and analyzed their differential expression in biotic and abiotic stresses finding one chitinase gene in particular ZmChi31 that is responsive to 11 different types of stresses.

1.     L22: RNA-seq analysis showed, this sentence should be changed to: Previous RNA-seq data analysis found…. The reason is that the authors did not perform the RNA-seq analysis in the current work.

2.     L37-39: GH18 are also found in bacteria.

3.     L41-45: I may suggest rewriting these sentences in order to explain in more detail the activity of endo and exo chitinases.

4.     L53-54: I suggest including the work conducted in transgenic lines of maize expressing different chitinases instead the work in other plant systems. i.e. doi: 10.1038/srep18067, https://doi.org/10.1016/j.aggene.2017.10.001

5.     L78-82: this justification needs to be rewritten since there is a very recent report of the genome-wide analysis of the maize chitinase gene family using the same version of the B73 maize genome (https://doi.org/10.3390/genes15081087)

6.     L117: In this section there is a lack of information about the phylogenetic analysis (i.e. which evolutionary model was used), and how many sequences were used from each plant genome? Did these sequences included all chitinase gene members of Arabidopsis and rice?

7.     L122: rewrite this sentence by including the name of the tool.

8.     L125-128: this paragraph explains biological concepts, I suggest to rewrite the paragraph and clearly explain collinearity in a synthesized form.

9.     L136: I suggest modifying the sub title to “Cis-acting  elements and functional analyses of maize chitinase genes”

10.  L112 & L171: Although WoLF PSORT is a good predictor to obtain subcellular location, my suggestion is to combine it with other subcellular location tools to get a more accurate protein subcellular location. This can be added as a supplementary table.

11.  L215-217 & L228: It Is not clear from the description of Figure 3 how the chitinase genes are sub-grouping, can they be correlated to chitinase Classes as in Figure 4, or according to GH families? Or are they only accommodated according to similarity of chitinases form the other two plant species, could you describe this in the text, please? Figure 3. Rewrite the figure caption completing the information on the phylogenetic tree.

12.  L218-220 aren´t they all chitinase genes? Please rewrite these sentences to clarify this section.

13.  L250: Figure 4. Which type of GH family or chitinase classes can be assigned to groups I, IV, and V?

14.  L33: Please rewrite this sentence: “Nineteen genes, including ZmChi2, ZmChi4, ZmChi8, ZmChi10, ZmChi12, ZmChi18, ZmChi19, ZmChi21, ZmChi23, ZmChi24, ZmChi25, ZmChi27, ZmChi32, ZmChi33, ZmChi34, ZmChi36, ZmChi37, ZmChi38, and ZmChi40, were not expressed in most maize tissuesbut, ZmChi2, ZmChi12, ZmChi18, ZmChi23, ZmChi24, ZmChi32, ZmChi33, ZmChi34 and ZmChi38 were not expressed in any tissue.

15.  L333: ZmChi28 is expressed slightly in Stem and SAM and almost undetectable in root and not only in 24H_Germinating Seed

16.  L338: Figure 8. Please, re-write the figure caption adding more information, such as in Figure 9, including what the abbreviations mean.

17.  L389. Figure 9A. Please write what NT, ELT, MLT, HT, LT, MHT and EHT means in the figure caption. Explain what the gray panels mean at the beginning of the figure footnotes.

18.  L441-446: It may be quite interesting to provide a more detailed explanation about the comparison of the expressed genes shared by both genotypes instead of comparing them individually. Please rephrase this paragraph.

19.  L:513: Why did you decide to compare only ZmChi5 and ZmChi9? The ideas are poorly explained when menctioning: “ZmChi5 and ZmChi9 were not observed under various types of abiotic and biotic stress” and “Zmchi5 was the most differentially expressed gene”. Please try to perform a more in-depth analysis of the comparison of the genes expressed under abiotic and biotic stresses and provide the reason to choose ZmChi5 and ZmChi9 for comparison.

20.  L515-517 The first of these two sentences is repetitive, the second sentence may be more adequate for the discussion section. I suggest deleting or rewriting this final comment of the results section.

21.  L519: Figure 11. “Gibberella stalk rot and salt” are in the wrong stress type, and in the figure caption it was written “blue indicates both increased and decreased expression” did you mean yellow?

22.  Discussion section: In general the discussion in different sections (L538-596, L625-683) seems more like a summary of results than a discussion.The discussion of chitinase gene expression under biotic and abiotic stress conditions is very vague. There is no discussion about the possible effect of some types of abiotic stress on the maize cell that could lead to the induction and/or repression of chitinase genes.

23.  L675: a broader discussion on ZmChi31 is required after your observations. Complement the discussion section describing any important  molecular information, if there is any silencing or overexpression studies conducted in this or its orthologous genes, or  is there any biochemical characterization conducted?

Comments on the Quality of English Language

Please check the manuscript again for any minor grammar English mistakes, or typos.

Reviewer 2 Report

Comments and Suggestions for Authors

Dear authors,

this study is very interesting with scientific and practical importance. The manuscript is about the genome-wide identification of the maize chitinase gene family and analysis of its response to biotic and abiotic stresses. The authors identified 43 chitinase genes in maize (Zea mays L.), grouped into six subfamilies with uneven chromosome distribution. The results of this study provide a new perspective for understanding the role of chitinase genes in the response to biotic and abiotic stresses in plants, and provide a theoretical basis for analysing the molecular mechanisms of maize adaptation to these stresses.

In the manuscript, introduction section is very well and clear written. The materials and methods section is very well written and given in details. The results presented in 11 figures and 4 supplementary tables are relevant to the proposed study. The discussion is very detailed and appropriate in the context of the results. The conclusions are supported by the results. The references are appropriate in the field of study. Among references there are a lot of recent references.

Before accepting of the manuscript, following parts have to be corrected:

7-14     Affiliations have to be in accordance with the Instructions for Authors.

37        [9, 10]     >      [9,10]       without space between brackets in the entire manuscript

64        Zea mays L.    >          Zea mays L.

89        The subsection titles in the entire manuscript have to be in accordance with the Instructions for Authors.

94,118,119      Arabidopsis thaliana   >   Arabidopsis thaliana        in the entire manuscript

106      methods          >          Methods

123      using   >          delete

309      and rice           >          ?          check it

310      Specifically, all Ks values were less than 1  >          check it

326      ZmChi1           >          ZmChi1

393      very low          >          extremely low            (ELT on the chart)

            Figure 9A       >          orders of the various temperature stress conditions on the charts have to follow order in the Figure 9A description

            Figure 9C        >          orders of the different sample types on the charts have to follow order in the Figure 9C description

402      t-salt    >          T-salt

403      s-salt    >          S-salt

404      t-ct, and s-ct    >          T-CT, and S-CT

406      snl       >          SNL

407      sdt       >          SDT

            srnl      >          SRNL

415      Schwabe          >          Schwabe

494      Figuer  >          Figure

518      Gibberella stalk rot     >          not abiotic stress

            Salt                             >          not biotic stress

521      blue indicates both     >          blue not exist on the chat

599      Fusarium graminearum          >          Fusarium graminearum

600      Fusarium         >          Fusarium

601      Leymus chinensis       >          Leymus chinensis

607      Serratia marcescens    >          Serratia marcescens

608      B. cinerea        >          Botrytis cinerea

609      Spodoptera frugiperda           >          Spodoptera frugiperda

615      T.harzianum    >          Trichoderma harzianum

709,710,711    Zou Kunliang  >          Kunliang Zou

728-995          The references have to be in accordance with the Instructions for Authors.

Reviewer 3 Report

Comments and Suggestions for Authors

In this study a genome-wide identification of the maize chitinase gene family and an analysis of its response to biotic and abiotic stresses were carried out. The maize B73_V5 genome was utilized for gene expression analysis.

The adopted methods are innovative and consistent with the aims of the research.

Results could constitute a great reference point for future research in the field.

The conclusions are coherent with results.

However, if authors took into account the following suggestions, the quality of the work could further be improved:

              The ‘Abstract’ contains an unclear sentence. Perhaps some words have been omitted. Lines 22-25.

              In the ‘Introduction’ section:

                    Some other recent studies about genes involved in abiotic stress response in maize could be mentioned, such as the work available at https://www.ncbi.nlm.nih.gov/pmc/articles/PMC10753923/

                    Similarly, references to very recent studies about the Genome-Wide Identification of a Maize Chitinase Gene Family could be added, such as  https://www.mdpi.com/2073-4425/15/8/1087

              In the ‘Results’ section:

                    Please check the heatmaps. For instance, the heatmap of figure 11 is wrong. Salt was included among biotic stress and Gibberella stalk root among abiotic stress.

                    Please make some figures more readable - for instance, figure 5.

              In the ‘Discussion’ section:

                    A figure/schema summarizing the more significant genes identified for each type of maize tissue and for abiotic and biotic stresses is recommended.

              In the ‘Conclusions’ section:

                    It could be appropriate to better underline the added value of the work compared to the existing studies.

A carefully rereading of the manuscript is strongly recommended.

Some minor issues:

              Please check the Italic form of scientific plant names – Lines 64, 473, 599, 607, 615

              Please change T.harzianum  into Trichoderma harzianum – Line 615

              Please check the format of the references - https://www.mdpi.com/journal/genes/instructions

Comments on the Quality of English Language

Minor editing of English language required.
